# Pharmacy Education and Clinical Pharmacy Training in France

**DOI:** 10.3390/pharmacy12060161

**Published:** 2024-10-29

**Authors:** Florence Ranchon, Sébastien Chanoine, Antoine Dupuis, Gaël Grimandi, Michel Sève, Stéphane Honoré, Benoît Allenet, Pierrick Bedouch

**Affiliations:** 1Université Lyon 1, Faculté de Pharmacie de Lyon, F-69008 Lyon, France; 2Hospices Civils de Lyon, Unité de Pharmacie Clinique oncologique, F-69495 Pierre-Benite, France; 3Université Grenoble Alpes, CNRS, UMR 5525, VetAgro Sup, Grenoble INP, CHU Grenoble Alpes, TIMC, UMR5525, Pharmacy Department, F-38043 Grenoble, France; schanoine@chu-grenoble.fr (S.C.); ballenet@chu-grenoble.fr (B.A.); pbedouch@chu-grenoble.fr (P.B.); 4ANEPC (National Association of Clinical Pharmacy Professors/Association Nationale des Enseignants de Pharmacie Clinique), F-38043 Grenoble, France; 5CPOPH (Conseil National Professionnel de la Pharmacie d’Officine et de la Pharmacie Hospitalière), National Professional Council for Community and Hospital Pharmacy, Université de Poitiers, Faculté de Pharmacie, F-75000 Paris, France; antoine.dupuis@univ-poitiers.fr; 6Conférence des Doyens de Faculté de Pharmacie Conference of Pharmacy Faculties Deans, Faculté de Pharmacie, Université de Nantes, F-75000 Paris, France; gael.grimandi@univ-nantes.fr; 7CIDPHARMEF (Conférence Internationale des Doyens des Facultés de Pharmacie d’Expression Française), Faculté de Pharmacie, Université Grenoble Alpes, F-38041 Grenoble, France; michel.seve@univ-grenoble-alpes.fr; 8Société Française de Pharmacie Clinique (SFPC), Faculté de Pharmacie, Aix-Marseille Université, F-13000 Marseille, France; stephane.honore@ap-hm.fr; 9Conseil Scientifique de la SFPC (Scientific Advisory Board), F-38043 Grenoble, France

**Keywords:** pharmacy education, clinical pharmacy, France, pedagogy

## Abstract

Clinical pharmacy education varies widely between European countries, and several major changes have taken place in France. This review aims to describe the current state of pharmacy education in France, focusing on clinical pharmacy. Research into legislative texts on pharmacy education in France was conducted based on the national database “legifrance”. A complementary search on clinical pharmacy teaching methods used in France was carried out on the Medline, Embase, Pascal and Francis database for articles published from 2008 to 30 April 2021. Pharmacy studies are taught in universities and last from six to ten years, depending on the student’s chosen options. The scientific curriculum is defined at the national level. Students choose their professional path after the fourth year with specialized courses. Whatever the direction chosen, all students have several internships, including a half-time one-year hospital internship, with patient-centered hospital functions within medical and pharmaceutical teams. The status of clinical pharmacy has been enhanced under French law and regulations, improving clinical pharmacy education, which is now skill-based, in a progressive, active, and dynamic process, with community or hospital pharmacists as university teachers and closer to real-life clinical pharmacy. Teaching is increasingly innovative, and this needs to be shared and reported in the literature. Several important reforms have modernized French pharmacy studies in recent years, conferring a pivotal place for clinical pharmacy.

## 1. Introduction

Based on the North American model of pharmacy studies initiated in the 1960s, European countries have upgraded pharmacy studies, shifting the paradigm from “product orientation” to “patient orientation” [1]. Clinical pharmacy emerged and was defined by the European Society of Clinical Pharmacy as a scientific discipline and a branch of pharmacy practice that aims to optimize the therapeutic use of medicines by patients and professionals in order to maximize the likelihood of achieving an optimal balance of clinical, humanistic, and economic outcomes [2]. At the same time, the practice of clinical pharmacy has expanded at both the community and hospital level [3], as encouraged in France by the French College of Clinical Pharmacy Teachers (ANEPC: *Association Nationale des Enseignants de Pharmacie Clinique*, https://www.anepc.fr/ accessed on 23 October 2024), the French Clinical Pharmacy Society (SFPC: Société Française de Pharmacie Clinique, https://sfpc.eu/ accessed on 23 October 2024), the European Society of Clinical Pharmacy (ESCP, escpweb.org accessed on 23 October 2024), and the recent inclusion of these activities in French regulations [4,5]. In this changing landscape, the Doctor of Pharmacy (PharmD) degree and specialist hospital pharmacist programs are adapting to ensure that pharmacists are well prepared to meet current health care needs.

In 1999, the European ministers of education signed the Bologna Declaration, fostering pharmacists’ mobility among European countries without requiring further training or validation. The European pharmacy degree is based on a two-cycle (i.e., bachelor’s and master’s) degree system with at least five years of study corresponding to the 300 European Credit Transfer and Accumulation System [6]. This training includes at least four years of full-time theoretical and practical training at a university or an equivalent accredited institute and at least six months of university-supervised practical training as part of a rotation between a community pharmacy and a hospital [7]. In 2016, Nunes-da-Cunha et al. highlighted that European Pharmacy education was still very much focusing on chemical sciences, whereas the United States had a higher load with the person-centered care approach [7]. More recently, the substantial variability in clinical pharmacy education between European countries has been further described [8,9]. As several major changes have occurred in France, this narrative study aims to describe the current state of pharmacy education in France, focusing on clinical pharmacy.

## 2. Materials and Methods

Two research strategies were implemented in this narrative study. The first described pharmacy education and more specifically clinical pharmacy education in France, delving into legislative texts based on “legifrance” (the national database: https://www.legifrance.gouv.fr/ accessed on 11 January 2024) until December 2023 using the keywords “pharmacy” and “education”. It was analyzed by a single author (FR). The second search focused on clinical pharmacy teaching methods used in France. This complementary search was realized on the Medline, Embase, Pascal and Francis database for articles published between 2008 and 30 April 2021, using the keywords “teaching methods”, “pedagogy”, “clinical pharmacy”, “pharmacy students”, “pharmacy education” and “France”. To be eligible for inclusion, studies had to be published in English or French and relate to the teaching methods used in France for clinical pharmacy education. FR and SC checked all published articles retrieved. A manual review of all selected articles reference lists was performed to identify any other relevant studies. The full text of all the eligible articles was screened, and the following criteria collected: date of publication; type of teaching method (simulation, problem-based learning, etc.); public; faculty involved; evaluation methods; and results.

## 3. Results

### 3.1. Current French Pharmacy Education

In France, pharmacy education is organized by the French Ministry of Health and the Ministry of Higher Education and Research and implemented in one of the twenty-four pharmacy faculties. It is open to all holders of the French *baccalauréat* school-leaving certificate. To be consistent with the European formula, a bachelor’s degree (1st, 2nd and 3rd years of pharmacy studies [10]) and a master’s degree (4th and 5th years [11]) are provided. The scientific training program is defined at the national level and each year includes a mandatory common core, accounting for 80–90% of ECTS. The first cycle of pharmacy studies is general training in the pharmaceutical sciences, with fundamental scientific knowledge in biological sciences (cellular and molecular biology, genetics, biochemistry, and physiology among others), pharmaceutical sciences (pharmacology, drug development, clinical pharmacy, etc.), chemistry (analytical, organic, mineral, etc.), and public health (prevention, intervention, and educational strategies for individuals and communities) [10]. The second cycle leads to an advanced degree in therapeutics and other health products, biological sciences, and public health [11]. The 4th year is critical, because students must choose a professional course: community pharmacy, pharmaceutical industry, and research or hospital practice (i.e., residency for medical biology and hospital pharmacy). The 4th year ends with the pharmaceutical certificate examination that covers all of the basic common training using case studies. During the 5th year, students complete a mandatory half-time one-year hospital internship called the university–hospital year (UHY). The third cycle begins in the 6th year. Students can choose between a short course (cycle) for general pharmacy and a long one in a hospital environment, leading to a specialist qualification as a medical biologist or hospital pharmacist. During the 6th year (or at the latest two years after the validation of the short third cycle), or later for long-cycle training (before the end of the advanced phase of residency; Figure 1), a doctoral thesis leads to the Doctor of Pharmacy degree (PharmD). The short cycle concerns community pharmacy, the pharmaceutical industry, or research. Students acquire in-depth knowledge and skills relating to their professional orientation and prepare their thesis. It includes a six-month full-time internship (community pharmacy or industry depending on the student orientation). For the research orientation, students prepare a PhD for at least three additional years. Long-cycle residency follows a national competitive examination at the start of the 5th year. Based on their ranking, students can choose between two options: hospital pharmacy or medical biology. The four- to five-year salaried residency comprises theoretical learning and six-month rotations in hospital, industry, and research units or health agencies, depending on the student’s choice and ranking in the national examination (Figure 1). Since 2017, the long third cycle has comprised three phases of progressive acquisition of knowledge, skills, and autonomy: 1/the “foundation phase”, lasting two years; 2/the “advanced phase”, lasting one year, or two years for the radiopharmacy option; and 3/the “consolidation phase”, lasting one year [12].

According to their chosen options, pharmacists may practice in community dispensing or hospital pharmacies, medical analysis laboratories, pharmaceutical or other industries (medical devices, cosmetics, agri-food industry, etc.), health authorities, universities, and research organizations.

### 3.2. Clinical Pharmacy Training in France

#### 3.2.1. Reinforcement of Clinical Pharmacy in Regulatory and Legislative Texts

In 2016, clinical pharmacy was defined in law as a mission of hospital pharmacies [4] and various types of services were specified in 2019 [5]: clinical pharmaceutical expertise, medication reviews, personalized pharmaceutical care plans, pharmaceutical interviews, and therapeutic patient education (Table 1). For community pharmacists, in addition to mandatory prescription analysis ahead of drug dispensing, new activities reinforcing the role of pharmaceutical care [13] have emerged: pharmaceutical interviews with patients receiving oral anticoagulants (2013), anti-asthmatic drugs (2014) or anticancer agents (2021), medication reviews for the elderly (2018), and vaccination (2019). In addition, coordinated community healthcare, which involves new roles for pharmacists, is being encouraged, enabling pharmacists to dispense treatment without medical prescription according to standardized protocols for a number of specified pathologies [14]. In 2018, the French Clinical Pharmacy Society (SFPC), created in 1984, proposed a new model of French clinical pharmacy services comprising three types of services: dispensing, medication review, and personalized pharmaceutical plans (Table 1). There is no discrepancy between broadening the array of clinical activities and the points identified in curricula. All these developments are concomitant with European recognition of the clinical pharmacy discipline, with the 2018 definition of clinical pharmacy by the European Society of Clinical Pharmacy.

This dynamic model highlights the adaptability of clinical pharmacy practices facing the risks associated with patients and treatments and enables practices to evolve from general to expert practice in primary care and in hospitals [15,16].

#### 3.2.2. Description of Clinical Pharmacy Teaching

The College of Clinical Pharmacy Teachers (*Collège des Enseignants de Pharmacie Clinique*, ANEPC), also founded in 1984, represents clinical pharmacy university teachers before the university and health authorities. Since 2008, pharmacists, like physicians, have been authorized to combine the functions of teacher and hospital practitioner, thereby professionalizing pharmacy studies [20] and minimizing the gap between studies and actual real-life practice, which is particularly important for clinical pharmacy.

Clinical pharmacy education is skills-based, in a progressive, active and dynamic process, with several clinical internships, as they are essential for students to become effective clinical pharmacists, as recommended by the Accreditation Council for Pharmacy Education and the American College of Clinical Pharmacy [21]. In France, there is as yet no formalized accreditation of training or teaching methods. The development of teaching methods is the responsibility of the national conference of deans of pharmacy faculties and the French College of Clinical Pharmacy Teachers, whose mission, as a college of the clinical pharmacy discipline, is to develop clinical pharmacy teaching.

##### Clinical Internships

The discipline of clinical pharmacy was introduced in French pharmacy education through the Bohuon reform, with a six-week hospital internship (1978), and then, above all, by the Laustriat–Puisieux reform (1985), creating a one-year half-time hospital internship during the fifth year of pharmacy studies: the UHY [22]. Since then, the various above-mentioned reforms have introduced other experiments in pharmacy studies second, third and fourth years, and in the sixth year for the community pharmacy option (Figure 1). The community pharmacy internship in the 2nd year lasts four weeks and aims to introduce students to the community pharmacist profession. In the 3rd and 4th years, two hospital or community pharmacy internships, lasting at least one week each, have been set up to apply theoretical learning in real-life situations following coordinated teaching on pathologies and therapeutics. During the fifth year, the UHY aims to develop clinical pharmacy skills by familiarizing students with safety and efficacy related to health products, biological examinations, medication-related problems, and therapy optimization at each stage of the care pathway, in collaboration with other health professionals, patients and caregivers. Students should be able to perform drug reconciliation, participate in educational sessions with patients, and draw up a medication plan. The 6th year, community pharmacy internship, is a professional internship lasting six months, reserved for students taking this option.

##### Clinical Pharmacy Teaching

Clinical pharmacy education is systematic during the first and second cycles, regardless of future orientation (Table 1). The clinical pharmacy discipline requires comprehensive knowledge, especially of pharmacology, semiology, and clinical biology, assimilated progressively depending on the pathologies and therapeutic classes studied in other disciplines. Teaching is also based on methodological points: how to dispense drugs, how to perform drug reconciliation, how to educate patients, etc. Clinical pharmacy learning begins mainly in the 3rd year of pharmacy studies and includes knowledge and skills related to the proper use of medications, as well as the analysis and prevention of adverse drug events (Table 1). The first clinical pharmacy activity to be taught to students is drug dispensing: pharmaceutical analysis of the prescription and/or patient’s request; the preparation of the doses to be administered; and the provision of information and advice necessary for proper use [15]. Students’ performance in pharmaceutical analysis and patient advice gradually increases over their courses and internships and is evaluated several times a year, adapted to the nature of the learning and their experience, and ranging from a classic exam to the simulation of a clinical situation. In the second cycle of the degree program, training also aims at acquiring skills in listening to and educating patients and mastering communication techniques, which is essential for professional clinical pharmacy practice [11]. Training in patient therapeutic education consists of a certificate course of at least 20 h. The UHY plays a major role in developing clinical pharmacy skills. At the end of the second cycle, the student will be able to communicate with patients and other health professionals, to perform medication reconciliation and medication review, thanks to sound knowledge of the main diseases, as well as clinical and biological investigations and related treatments, besides participating in patients’ therapeutic education [11]. The third cycle enhances clinical pharmacy skills. During the residency in the hospital pharmacy option, a specialized clinical pharmacy course is mandatory during the foundation phase. It aims at a higher quality of specialized clinical pharmacy with concrete applications shared and discussed with hospital clinical pharmacy teachers. For the community pharmacy option, the 6th year focuses on specific populations particularly at risk of adverse events: children, the elderly, patients with cancer or kidney or liver failure, and pregnant or breastfeeding women. This training prepares for pharmaceutical interviews with patients treated with oral anticoagulants, anti-asthmatic drugs, or anticancer agents, together with shared medication reviews for the elderly.

##### Continuing Education in Clinical Pharmacy

In addition to personal lifelong continuous learning notably in pharmacology, to stay abreast of therapeutic advancements, ongoing training in clinical pharmacy is also available for residents or graduate pharmacists (mainly community or hospital pharmacists). Several continuing education programs are on offer at French faculties, including university diploma courses (Figure 2). These are specialization certificates, set up by the universities themselves, and therefore do not have a national character. A specialized clinical pharmacy master’s degree is now available in France for the first time. This was initiated in 2023 and offers the possibility of specialization in clinical pharmacy. All these training courses remain optional. Pharmacists have no obligation to validate them to practice clinical pharmacy.

#### 3.2.3. Teaching Methods in Clinical Pharmacy Education

To date, there are no guidelines on teaching techniques in clinical pharmacy training in France. Coordinated teaching (i.e., transverse courses with the involvement of teachers from different disciplines) is clearly mentioned in French law, as well as the duration of experiential internships in hospital or community pharmacy, with at least a six-month full-time equivalent at the end of the second cycle [11]. Teaching methods in clinical pharmacy are consequently chosen by the individual teacher, depending on their individual possibilities, training, availability, etc., and therefore differ according to faculties. In 2008, Planus et al. advocated that considerable effort should be made to offer validated learning methods based on active pedagogy in French pharmacy faculties [23]. Although the most widely used approach in pharmaceutical studies is still transmissive teaching, in which students are basically passive [24], several innovative teaching methodologies have been introduced, facilitated by digital technology. Some French pharmacy faculties have developed experimental pharmacies to train future pharmacists, using simulation, work situations, and role-play. “Serious games”, which aim to teach students actively by forcing them to make decisions in contexts that are close to real-life situations in a protected environment [25], are beginning to be used in French clinical pharmacy studies. Two serious games, PROFFIteROLE (namely Pratiques Officinales et Jeux de roles: http://klip.univ-lille.fr/fiche/40-pratiques-officinales-et-jeux-de-role-proffiterole accessed on 24 October 2024) and MISSION OFFI’SIM (https://pharma.univ-lorraine.fr/innovations-pedagogiques/ accessed on 24 October 2024), were developed, respectively, by the Lille and Nancy faculties, for fifth- and sixth-year students to train in community pharmacy practices.

A search for articles published from 2008 to 30 April 2021 on clinical pharmacy teaching methods used in France retrieved six methods (Table 2) out of 694 articles in total.

In 2010, Roustit et al. implemented an Internet problem-based learning tool for clinical pharmacists and pharmacy students [26]. Barbier et al. focused on anticoagulant pharmaceutical interviews and proposed blending e-learning, role-play, and simulation [27,28]. Lawson et al. also described blended learning associated with the peer evaluation of knowledge of adverse drug reactions and drug–drug interaction management. Students were mainly positive about this approach’s potential in terms of learning outcomes [29]. Encouraging experiments with flipped classrooms were reported in 2021 by Lescuyer et al. for pediatric clinical pharmacy learning with videos on pediatric medication and fever [30]. Participants (sixth-year pharmacy students, residents, and pharmacists) had to summarize the most important notions and prepare clinical cases. This new tool was tested on 116 students, 95.7% of whom were satisfied with this new teaching method [22]. In addition, patient involvement in pharmacy studies is a complementary teaching method. At Strasbourg University, Schraub et al. reported their experience and evaluation of patient partners in teaching medical students (physicians and pharmacists) in oncology [31]. They highlighted that patient partners provided complementary information that students found useful for their future practice. To promote this practice, the Grenoble medicine and pharmacy faculties created a University Patients and Caregivers Department (May 2021), allowing tight partnership between caregivers and patients in caregivers’ initial training. Finally, in 2017, Gaboreau et al. listed all interprofessional courses in France, bringing together general practitioners and community pharmacy students (fifth or sixth years of pharmacy studies). They showed that such courses were gradually being introduced in medical and pharmacy curricula in France, with 10 universities (out of the 24 pharmacy and 36 medicine faculties) providing interprofessional courses dealing with students’ representations of each profession and the identification of respective competence fields and complementarity points [32]. This collaborative patient-centered approach is essential for healthcare students’ training and should strengthen all caregivers’ professional identity and missions [33]. All these experiences highlight the drive to modernize pharmacy studies, with greater participatory student involvement based on active teaching methods.

## 4. Discussion

French pharmacy education was already described in 2007 [34] and 2008 [35], but evolutions and changes have been so dramatic that this paper has proved necessary, especially since the European framework has also evolved and proposed some harmonization elements among European countries [6]. However, the recent survey collecting information on pharmacy education in Europe between October 2018 and January 2019 highlights great variability in education and practice patterns [8]. A more precise evaluation, beyond the theoretical program, of the number of hours of clinical pharmacy teaching in each French faculty is still difficult to obtain. In the European study, this point was assessed to be 15 h of clinical pharmacy teaching per semester, which represents the lowest number of hours among the European countries that responded to the survey in 2018 [8]. Implementing common training quality standards for clinical pharmacy education across Europe is still needed.

Clinical pharmacy teaching is still facing the challenge of reducing the gap between academic education and real-life practices so as to better prepare pharmacy students in a context of educational reform and practice changes [36], besides meeting patient needs. New clinical pharmacy activities, especially in community pharmacy, require new skills, which future pharmacists must acquire at the end of their training, reinforcing the need for tailored initial and continuous education [36], along with multidisciplinary teaching, especially for future general practitioners [37]. In addition, digital technologies, AI-powered apps, and tools for clinical pharmacy services are booming [38]. Training on how to use these tools is essential for good data-driven decision-making and for optimizing medication management for enhanced patient outcomes [39].

We highlight four important points worth promoting and developing for clinical pharmacy teachers and state decision-makers.

-First, pharmacy students need to receive broader training in ethics and health education to foster a personalized approach based on the patient rather than on the medicine [40] by integrating a preventive approach. Although training in communication and soft skills is now an integral part of French pharmacy studies, the COVID-19 pandemic has caused the general public to mistrust drugs and pharmaceutical companies and has been a reminder of the importance of understanding patients’ socio-behavioral features when using medication to achieve optimal clinical and humanistic outcomes. Social pharmacy needs further developing in pharmacy studies, as the interaction between patients and pharmacists is increasing, notably with clinical pharmacy activities (such as drug reconciliation, pharmaceutical interviews, and therapeutic education). State decision-makers must display clear public health priorities, with the use of human and material resources to achieve these objectives.-Secondly, as in United States [41], declining interest in pharmacy as a career has been observed in France. One of the counteracting levers rests on developing a strong professional identity, as also highlighted in a French context [33]. Recently, the 2023–2024 report by the Academic Affairs Standing Committee, published in the American Journal of Pharmaceutical Education, stresses the urge for pharmacy education to prioritize enhancing a stronger professional identity so as to develop collaboration among pharmacists [39]. The major issue that could change clinical pharmacist practice is to get the pharmaceutical profession to appropriate a caregiver identity [33]. This point must integrate interprofessional education, which is a prerequisite to building a collaborative practice environment and optimizing patient care [42]. Clear guidelines must be proposed in the French faculties of Health to implement this approach more widely.-Thirdly, as already mentioned by Planus et al., in 2008, few French researchers have described teaching strategies in pharmacy training [23]. Just as evidence-based medicine is taught to pharmacy students, evidence-based pharmacy education must be developed by pharmacy teachers. Teachers’ involvement in educational innovation should be promoted [24]. Clinical pharmacy training needs to use a combination of teaching techniques, including e-teaching, and to prioritize one-on-one and one-site teaching, using simulated or real-life professional situations to work on soft skills (Figure 3). To develop these approaches, the number of clinical pharmacy teachers and their qualifications in pedagogical methods must be increased.-Fourthly, student assessment must be suited to teaching methods and skill acquisition and based on objective learning progress assessment [23]. In clinical pharmacy, examinations should allow teachers to assess students’ clinical knowledge, communication, and problem-solving skills. Objective structured clinical examination formats, gold standards for evaluating clinical skills in medicine [43], and examinations in real-life professional situations [44] should probably be more widely used by French teachers. Pharmacy faculty members therefore need to increase training in the proper design of educational research [45]. One of the digital technology opportunities is that it potentially makes it easier to expand exchanges and bring together French-speaking teachers and students around clinical pharmacy training and educational research. Belgian and Swiss pharmacy teachers recently published an open randomized controlled study comparing an online text-based scenario and a “serious game” to teach triage in the case of coughing. They showed that an online lesson, based on a case study, can be introduced in different countries with only minor changes (e.g., adapting the local drug names) [25]. The French-speaking world offers a potential to develop interrelations and enrich clinical pharmacy education and research.

## 5. Conclusions

In recent years, many fundamental reforms have modernized French pharmacy studies, giving a significant place to clinical pharmacy via several internships. The fast-changing professional pharmacy context requires regularly adapting training programs. Granted, more and more clinical pharmacy courses are developing using active teaching methods, but their implementation is being held back by classroom demographics and the shortage of the resources necessary for faster development.

## Figures and Tables

**Figure 1 pharmacy-12-00161-f001:**
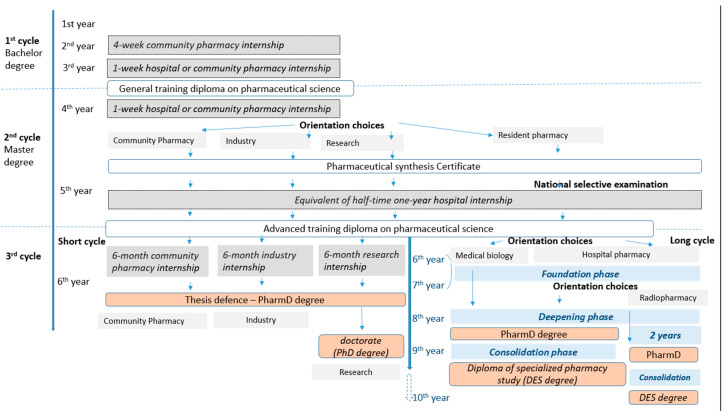
The French Pharmacy Education system. DES degree: Diploma of specialized pharmacy study (*Diplôme d’Etudes Spécialisées)*; Pharm D: Pharmacy Doctor.

**Figure 2 pharmacy-12-00161-f002:**
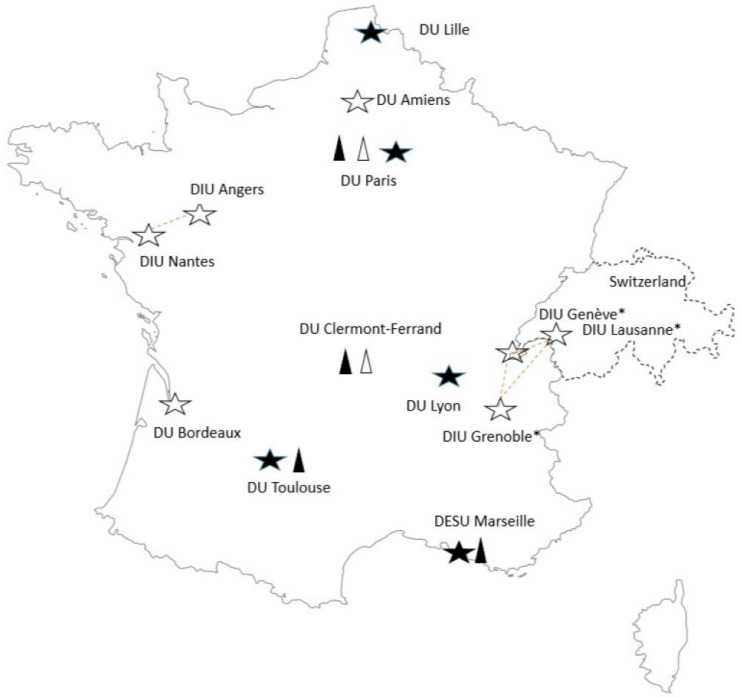
University diplomas in clinical pharmacy available in France in 2023. White star: clinical pharmacy; dark star: oncology clinical pharmacy; white triangle: clinical pharmacy in pediatry; dark triangle: clinical pharmacy in geriatry. University diplomas (*DIU*: *diplôme inter-universitaire, DU: diplôme d’université, DESU: diplôme d’études supérieures universitaires*) * in partnership with the University of Geneva and the University of Lausanne, Switzerland (Certificate of Advance Studies in Clinical Pharmacy).

**Figure 3 pharmacy-12-00161-f003:**
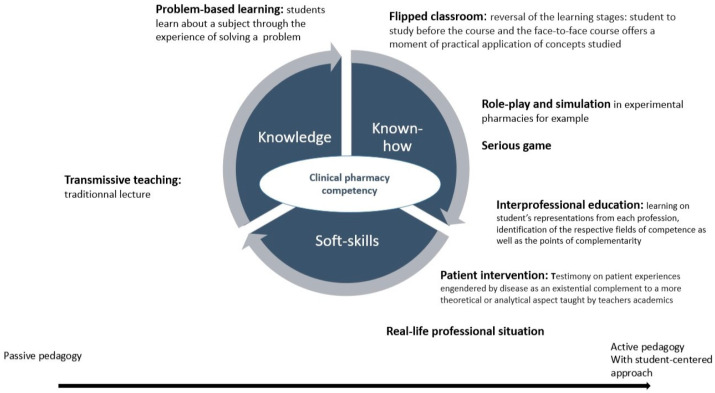
Combination of pedagogical techniques in a competency-based approach for teaching clinical pharmacy.

**Table 1 pharmacy-12-00161-t001:** Clinical pharmacy formation during French pharmacy studies regarding Clinical Pharmacy Services (CPS) authorized in France.

CSP	Definition of CSP [15,16]	Year of Training	Items of Learnings Specify in the Law
Drug dispensing	It includes, chronologically, 1/pharmaceutical analysis of the medical prescription and/or the patient’s request; 2/preparation of the doses to be administered; 3/provision of information and advice necessary for the correct use of the health product [15].	1st cycle: 3rd year	Dispensing, drug traceability, pharmaceutical files, proper use of medications, analysis and prevention of adverse drug events (ADEs)
		2nd cycle: 4th year, community pharmacy (CP)	Dispensing of health products: analysis and validation of the prescription
		2nd cycle: 5th year	Preparatory education for taking up hospital activities during UHY
		2nd cycle: 5th year, CP	Dispensing of health products
		2nd cycle: 5th year, *internat*	Preparatory education for taking up as a pharmacy resident
		3rd cycle, *internat*: hospital pharmacy (HP)	Clinical pharmacy and therapeutic management
Clinical pharmaceutical expertise	In-depth analysis of the medico-pharmaceutical situation of the patient to propose an optimization of the therapeutic of the patient and his follow up [15].	1st cycle: 3rd year	Proper use of medications, analysis and prevention of ADE
		2nd cycle: 5th year	Preparatory education for taking up hospital activities during UHY
		2nd cycle: 5th year, CP	Pharmaceutical and biological monitoring of patients Comprehensive pharmaceutical care of the patient: self-medication, self-prescription, optional prescription drugs, advice
		2nd cycle: 5th year, *internat*	Preparatory education for taking up as a pharmacy resident
		3rd cycle: 6th year, CP	Optimization of outpatient pharmaceutical care, of aging, of pregnancy monitoring, support from birth to early childhood, and from cancer patients
		3rd cycle, *internat*: HP	Clinical pharmacy and therapeutic management
Medication review	Result of drug reconciliation (a formal process in which healthcare provider’s partner with patients and their families ensure accurate and complete medication information transfer at interfaces of care). This includes admission and discharge from a hospital or changes in care setting services [17]. Associated with a clinical pharmaceutical expertise.	2nd cycle: 5th year	Preparatory education for taking up hospital activities during UHY
		2nd cycle: 5th year, *internat*	Preparatory education for taking up as a pharmacy resident
		3rd cycle, *internat*: HP	Clinical pharmacy and therapeutic management
Shared medication reviews	They include 1/elderly patient interview to collect relevant information on treatment, biology, etc.; 2/In-depth analysis of the medico-pharmaceutical situation of the patient leading to pharmaceutical opinion with suggestions to patient’s doctor to revise patient’s treatment; 3/Advice to patients; 4/Adherence follow-up.	2nd cycle: 5th year, CP	Carry out the pharmaceutical follow up
		3rd cycle: 6th year CP	Learning on specific populations particularly at risk of ADE: elderly
Therapeutic patients education (TPE)	TPE helps patients acquire or maintain the skills they need to manage their life with a chronic disease in the best possible way [18]. It implies a multi-professional team.	2nd cycle: 4th year	TPE (2 ECTS): definition, principle of implementation
		2nd cycle: 5th year CP	TPE and patient support (2 ECTS)
		3rd cycle, *internat*: HP	Clinical pharmacy and therapeutic management (4 ECTS)
Pharmaceutical consultation	Exchange between a patient and a pharmacist to collect information and to reinforce advice, messages of prevention, and education.	2nd cycle: 5th year	Preparatory education for taking up hospital activities during UHY
		2nd cycle: 5th year, CP	Carry out the pharmaceutical follow up (2 ECTS)
		3rd cycle, 6th year CP	Learning on specific populations particularly at risk of ADE
Vaccination	Vaccination against the seasonal influenza virus and COVID 19 [19].	2nd cycle: 5th year	Specialized activities at community pharmacy (3 ECTS): training in the vaccine procedure

ADE: adverse drug events, CP: community pharmacy, CSP: Clinical pharmacy services, HP: hospital pharmacy, UHY: university-hospital year, TPE: Therapeutic patient education.

**Table 2 pharmacy-12-00161-t002:** French research on clinical pharmacy teaching.

Reference	Teaching Method	Details of the Pedagogic Technic	Topics	Public	Objectives	Methods of Evaluation	Results of Evaluation
Schraub S et al., 2021, Strasbourg, France [23]	Patient-partner	Patients explain their diagnostic, therapeutic journey, feelings about disease and relationship with their oncologists Discussion with students	Cancer	187 medicine and 131 pharmacy students	To complete students formation with real existential experiences	Questionnaires for students, patients teachers and moderator To assess the expected benefits at the end of the training, at 6-month	- Meeting students expectations in 98% - 1/3 of students were destabilized by this training - Great satisfaction of patients teacher - 6 months later, 30% of student respondents said that these testimonies had or could have an impact on their practices
Lescuyer C. et al., 2021, Paris Descartes, France [22]	Flipped classroom	4 videos, lasting between 4 and 8 min, on pediatric medication and fever. Participants had to summarize the important notions and prepare clinical cases	Pediatric clinical pharmacy	86 pharmacy students on 6th year, 29 residents, and 1 community pharmacists	To develop a participative learning	Satisfaction questionnaires	92% were satisfied on the vision of the video before the course, 81% appreciated this upstream work for the summary, and finally 95.7% were satisfied on this new teaching method
Lawson R. et al., 2019, Limoges, France [21]	Blended learning Peer evaluation	The teaching used 3 steps 1/ Pharmacology courses on adverse drug reaction 2/ Distance working: groups of 4 students have to work on clinical cases and to submit individually to the teacher a structured report on the case for peer evaluation. Peers’ work was performed anonymously. 3/ Classroom based activity: to improve their work collectively and an oral group presentation on clinical cases	Drug-drug interaction	72 3rd year pharmacy students	To better train pharmacy students in drug-drug interactions management	Analysis of student perceptions via an online survey using Google Forms^®^ platform.	In a scale ranging from 0 to 10, the overall feeling about this activity was ranked 8.0 ± 1.0. Benefits from this teaching approach were mostly identified: helpful in memorizing knowledge (39.7%, n = 27); a better understanding of knowledge (55.9%, n = 38); questioning the students own work (64.7%, n = 44); acquiring additional knowledge (70.6%, n = 48) and critical thinking skills (72.1%, n = 49).
Barbier et al., 2014, Paris Sud and Béclère hospital pharmacy [19]	Blended learning	1/ An e-learning module including steps of knowledge assessments and theoretical slideshow with sounded comments. The student manages his training at his own pace, he can go back at every step and 2 ideal virtual consultation scenarios. 2/ A practical training with a real pharmaceutical consultation on VKA	VKA	14 pharmacy students	To provide an individualized training and to reduce the overall time necessary to formation	Satisfaction questionnaires	Qualitative feedbacks from learners showed good adhesion and also identified proposals to improve the module.
Barbier et al., 2013, Paris Sud and Béclère hospital pharmacy [20]	Simulation	The training course was divided into 5 phases: 1/an assessment of prior training knowledge; 2/ theoretical training that summarized the general context and key consultation points; 3/ simulated training consultations involving pretend patients and pharmacists; 4/ a reassessment of knowledge; 5/ 2 real-life consultations supervised by tutors.	VKA	34 pharmacy students	To develop a training program on anticoagulant consultation	Evaluation before/after teachings Final validation on real-life consultation Satisfaction questionnaires	The 2 knowledge assessment sessions reported significant individual progress. The pharmacy training was validated after 2 real-life evaluations. The satisfaction survey showed that 100% of learners were satisfied.
Roustit et al., 2010, Grenoble [18]	Problem based learning	An internet problem-based learning tool. All users must be able to provide clinical cases from their experience and work on their colleagues’ cases through a Web 2.0 tool.	Clinical pharmacy	Clinical pharmacists and pharmacy students	To improve training for pharmaceutical care providing	Not assessed	Not assessed

ANEPC: French College of Clinical Pharmacy Professors (ANEPC for *Association Nationale des Enseignants de Pharmacie Clinique*); VKA: vitamin K antagonist.

## Data Availability

All data are contained within the article.

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
