# Peer review of "Pharmacy Education and Clinical Pharmacy Training in France"

_pharmacy, 2024, doi:10.3390/pharmacy12060161_

Round 1

Reviewer 1 Report

Comments and Suggestions for Authors

Thank you for giving me the opportunity to review this work: “Pharmacy Education and Clinical Pharmacy Formation in France.” It is interesting to see the structure of pharmacy education, which is different from that in the US or the UK. However, the manuscript is a bit confusing and hard to follow. You could probably add another figure or table that explains the education part of the curricula, excluding the training. The introduction is quite concise and has only four references, which, in my opinion, need to be expanded. Perhaps a general background about education and information on how many colleges there are, as well as whether they all follow the same system, would be beneficial.

Please also have a look at this article and indicate whether your review is a narrative review, scoping review, or mini-review, and adjust accordingly.

https://www.emeraldgrouppublishing.com/archived/products/journals/call_for_papers.htm%3Fid%3D5730

Author Response

Thank you for giving me the opportunity to review this work: “Pharmacy Education and Clinical Pharmacy Formation in France.” It is interesting to see the structure of pharmacy education, which is different from that in the US or the UK.

Thank you very much for your time and appreciation of this manuscript.

However, the manuscript is a bit confusing and hard to follow. You could probably add another figure or table that explains the education part of the curricula, excluding the training. The introduction is quite concise and has only four references, which, in my opinion, need to be expanded. Perhaps a general background about education and information on how many colleges there are, as well as whether they all follow the same system, would be beneficial.

We agree with you. We have modified the introduction part as you suggested. You can see directly the revised Introduction in the manuscript.

Please also have a look at this article and indicate whether your review is a narrative review, scoping review, or mini-review, and adjust accordingly. https://www.emeraldgrouppublishing.com/archived/products/journals/call_for_papers.htm%3Fid%3D5730

Thank you for this reference. Our review is categorized as a narrative review.

Reviewer 2 Report

Comments and Suggestions for Authors

Introduction. The Introduction is comprehensive and very long. It is well done. There are several grammatical errors here and in the rest of the paper that require attention.

Material and Methods. Please expand this section to identify who was responsible for this research and how papers were chosen to be referenced. Also note how many papers were identified through the search technique and the criteria—other than date of publication— used to include them.

Results. Figure 1 (l. 102) presents a schematic covering all the possible pathways to practice in various environments. Acronyms are translated into French but not English. The figure is quite complex for those—like this reviewer—who operate in very different systems. The longest pathway (10 years) is not much different in length than a US track that would include a PharmD (3-4 years), a residency (1-2 years) and specialty training.  

Changing French laws/descriptions regarding the emergence of an expanded array of clinical activities are well described (ll. 113-126). Table 1 maps the expanded array of clinical activities to delivery points in curricula. It is a good exercise in avoiding gaps in achieving new practice goals. If there are still gaps, they are not noted.

Is Section 3.2.2.2., Theoretical clinical pharmacy education, perhaps misnamed? It seems that activities are the opposite of theoretical—they are practice-based, hands on activities. Assessment of the students’ performance is not addressed. Are there periodic assessment activities that students are ready for the next set of activities?

Section 3.2.2.3. Continuing education in clinical pharmacy is very brief. If the subject is outside the scope of the paper, that should be noted. If not, information about training the most highly trained clinical pharmacists should be expanded.

Section 3.2.3. Teaching methods in clinical pharmacy education is the strongest section of the paper. It is well referenced and presented. It appears that many of these methods are not yet widely adopted. There is no mention of accreditation requirements related to the teaching methods.

Discussion. The Discussion does a good job of outlining what needs to be done but does not describe barriers to achieving the goals outlined. A more in-depth analysis of how to achieve better clinical pharmacy education and training in French institutions is needed. Figure 3 has good content but the font is too small to read.

Conclusion. Some new concepts are introduced in the Discussion and Conclusion; e.g., understanding patients’ sociobehavioral aspects in the use of medication and soft skills. These are important and should first appear in the Introduction and be followed up in the Discussion and Conclusion. One of the most important thoughts in the Conclusion is not expressed as a complete sentence (ll. 295-296).

Overall, this is a strong effort to cover a huge topic. So much more could have been added comparing where the French system us relative to US and other educational/training programs. The areas of advanced clinical training (residencies and board certification) are not sufficiently addressed but may require another study.

Comments on the Quality of English Language

Many grammatical errors need attention.

Author Response

Introduction. The Introduction is comprehensive and very long. It is well done. There are several grammatical errors here and in the rest of the paper that require attention.

We are sorry for these issues, a native English speaker has checked the text.

Material and Methods. Please expand this section to identify who was responsible for this research and how papers were chosen to be referenced. Also note how many papers were identified through the search technique and the criteria—other than date of publication— used to include them.

Agreed and done. We specified all these points in the method section.

“Two research strategies were implemented in this narrative review. The first described pharmacy education, and more specifically clinical pharmacy education in France, delving into legislative texts based on “legifrance” (the national database: https://www.legifrance.gouv.fr/) until December 2023, using the keywords “pharmacy” and “education. It was analyzed by a single author (FR). The second search focused on clinical pharmacy teaching methods used in France. This complementary search was realized on the Medline, Embase, Pascal and Francis database for articles published between 2008 and April 30, 2021, using the keywords “teaching methods”, “pedagogy”, “clinical pharmacy”, “pharmacy students”, “pharmacy education” and “France”. To be eligible for inclusion, studies had to be published in English or French, and relate to the teaching methods used in France for clinical pharmacy education. FR and SC checked all published articles retrieved. A manual review of all selected articles reference lists was performed to identify any other relevant studies. The full text of all the eligible articles was screened and the following criteria collected: date of publication; type of teaching method (simulation, problem-based learning….); public, faculty involved, evaluation methods and results.”

Results. Figure 1 (l. 102) presents a schematic covering all the possible pathways to practice in various environments. Acronyms are translated into French but not English. The figure is quite complex for those—like this reviewer—who operate in very different systems. The longest pathway (10 years) is not much different in length than a US track that would include a PharmD (3-4 years), a residency (1-2 years) and specialty training. 

We corrected this figure 1 as proposed. We added color on the figure also to be more comprehensible.

Changing French laws/descriptions regarding the emergence of an expanded array of clinical activities are well described (ll. 113-126). Table 1 maps the expanded array of clinical activities to delivery points in curricula. It is a good exercise in avoiding gaps in achieving new practice goals. If there are still gaps, they are not noted.

Thank you for your comment. We added that no gaps between expanded array of clinical activities and points in curricula were identified.

“There is no discrepancy between broadening the array of clinical activities and the points identified in curricula.”

Is Section 3.2.2.2., Theoretical clinical pharmacy education, perhaps misnamed? It seems that activities are the opposite of theoretical—they are practice-based, hands on activities. Assessment of the students’ performance is not addressed. Are there periodic assessment activities that students are ready for the next set of activities?

We agree with you. We initially wanted to explain the conceptual framework to do clinical pharmacy activities but the terms “theoretical clinical pharmacy education” are probably confusing. We have simplified the expression with “clinical pharmacy teaching”.

Thank you for your insightful comment on student’s evaluation. We added this point on the manuscript.

Students’ performance in pharmaceutical analysis and patient advice gradually increases over their courses and internships and is evaluated several times a year, adapted to the nature of the learning and their experience and ranging from a classic exam to the simulation of a clinical situation.

Section 3.2.2.3. Continuing education in clinical pharmacy is very brief. If the subject is outside the scope of the paper, that should be noted. If not, information about training the most highly trained clinical pharmacists should be expanded.

We think this point is important. We have modified the manuscript as follow.

“In addition to personal lifelong continuous learning notably in pharmacology, to stay abreast of therapeutic advancements, ongoing training in clinical pharmacy is also available for residents or graduate pharmacists (mainly, community or hospital pharmacists). Several continuing education programs are on offer at French faculties, including university diploma courses (Figure 2). These are specialization certificates, set up by the universities themselves, and therefore do not have a national character. A specialized clinical pharmacy Master’s degrees is now available in France for the first time. It was initiated in 2023 and offers the possibility of specialization in clinical pharmacy. All these training courses remain optional. Pharmacists have no obligation to validate them to practice clinical pharmacy.”

Section 3.2.3. Teaching methods in clinical pharmacy education is the strongest section of the paper. It is well referenced and presented. It appears that many of these methods are not yet widely adopted. There is no mention of accreditation requirements related to the teaching methods.

Indeed, in France, there is not yet formalized accreditation of training or teaching methods. The development of teaching methods is the responsibility of the national conference of deans of pharmacies faculty and the French College of Clinical Pharmacy Teachers (ANEPC, for Association Nationale des Enseignants de Pharmacie Clinique, https://www.anepc.fr/), which as a college of the discipline of clinical pharmacy has the mission of developing clinical pharmacy teaching.

This point has been added in this part.

“In France, there is as yet no formalized accreditation of training or teaching methods. The development of teaching methods is the responsibility of the national conference of deans of pharmacy faculties and the French College of Clinical Pharmacy Teachers, whose mission, as a college of the clinical pharmacy discipline, is to develop clinical pharmacy teaching”

Discussion. The Discussion does a good job of outlining what needs to be done but does not describe barriers to achieving the goals outlined. A more in-depth analysis of how to achieve better clinical pharmacy education and training in French institutions is needed. Figure 3 has good content but the font is too small to read.

Thank you for your comment. We develop these points in the discussion section.

We have modified figure 3 to make it readable.

Conclusion. Some new concepts are introduced in the Discussion and Conclusion; e.g., understanding patients’ sociobehavioral aspects in the use of medication and soft skills. These are important and should first appear in the Introduction and be followed up in the Discussion and Conclusion. One of the most important thoughts in the Conclusion is not expressed as a complete sentence (ll. 295-296).

Sorry for this issue. We agree and we have moved this notion in the discussion section.

Overall, this is a strong effort to cover a huge topic. So much more could have been added comparing where the French system us relative to US and other educational/training programs. The areas of advanced clinical training (residencies and board certification) are not sufficiently addressed but may require another study.

Thank you very much for your time and appreciation of this manuscript.

Comments on the Quality of English Language. Many grammatical errors need attention.

We are sorry for these issues, a native English speaker has checked the text.

Reviewer 3 Report

Comments and Suggestions for Authors

The article is well written, but in my opinion, it would be interesting only for local readers. It describes only the education system in clinical pharmacy in France. The authors just describe the law acts and a few articles dedicated to teaching in France. It doesn't offer any practical recommendations or suggestions for teachers or academics. 

Author Response

The article is well written, but in my opinion, it would be interesting only for local readers. It describes only the education system in clinical pharmacy in France. The authors just describe the law acts and a few articles dedicated to teaching in France. It doesn't offer any practical recommendations or suggestions for teachers or academics. 

Thank you very much for your time and appreciation of this manuscript. As you suggested, we decided to add recommendations for academics teachers in the discussion section.

We think that it is interesting to share our methods of learning clinical pharmacy to improve them but also for other countries because the structure of pharmacy education differs from that in the US or the UK for example.  Other reviewer highlighted that this manuscript could be helpful for their reflection and practice of pharmacy education.

Reviewer 4 Report

Comments and Suggestions for Authors

The paper describes clinical pharmacy in France and also teaching methods in clinical pharmacy education. Although, a nice description I do not think that the paper is scientific enough and can not be qualified as a review. The methods section is only briefly described and lacks a systematic approach, No attempt is made to for example compare the situation in other countries or put the results in a broader perspective. 

Author Response

The paper describes clinical pharmacy in France and also teaching methods in clinical pharmacy education. Although, a nice description I do not think that the paper is scientific enough and can not be qualified as a review. The methods section is only briefly described and lacks a systematic approach, No attempt is made to for example compare the situation in other countries or put the results in a broader perspective. 

Thank you very much for your time and appreciation of this manuscript.

“Two research strategies were implemented in this narrative review. The first described pharmacy education, and more specifically clinical pharmacy education in France, delving into legislative texts based on “legifrance” (the national database: https://www.legifrance.gouv.fr/) until December 2023, using the keywords “pharmacy” and “education. It was analyzed by a single author (FR). The second search focused on clinical pharmacy teaching methods used in France. This complementary search was realized on the Medline, Embase, Pascal and Francis database for articles published between 2008 and April 30, 2021, using the keywords “teaching methods”, “pedagogy”, “clinical pharmacy”, “pharmacy students”, “pharmacy education” and “France”. To be eligible for inclusion, studies had to be published in English or French, and relate to the teaching methods used in France for clinical pharmacy education. FR and SC checked all published articles retrieved. A manual review of all selected articles reference lists was performed to identify any other relevant studies. The full text of all the eligible articles was screened and the following criteria collected: date of publication; type of teaching method (simulation, problem-based learning….); public, faculty involved, evaluation methods and results”

The discussion section was also profoundly modified as you suggested. You can see directly the revised Discussion in the manuscript.

Round 2

Reviewer 3 Report

Comments and Suggestions for Authors

The paper is well-written and focused on the local situation in France. The discussion section includes some recommendations and comments, making the paper more interesting for readers outside France. 

Author Response

Reviewer 3

The article is well written, but in my opinion, it would be interesting only for local readers. It describes only the education system in clinical pharmacy in France. The authors just describe the law acts and a few articles dedicated to teaching in France. It doesn't offer any practical recommendations or suggestions for teachers or academics. 

Thank you very much for your time and appreciation of this manuscript. We understand your comment but other countries have recently published their curricula in the journal Pharmacy, so we think that we are in the scope of this journal.

For example:

  • Sandulovici R, Mircioiu C, Rais C, Atkinson J. Pharmacy Practice and Education in Romania. Pharmacy (Basel). 2018 Jan 8;6(1):5. doi: 10.3390/pharmacy6010005. PMID: 29316686; PMCID: PMC5874544.
  • Božič B, Obreza A, Atkinson J. Pharmacy Practice and Education in Slovenia. Pharmacy (Basel). 2018 Dec 24;7(1):4. doi: 10.3390/pharmacy7010004. PMID: 30586866; PMCID: PMC6473832.
  • Muceniece R, Riekstina U, Maurina B, Enina V, Atkinson J. Pharmacy Practice and Education in Latvia. Pharmacy (Basel). 2018 Jan 20;6(1):9. doi: 10.3390/pharmacy6010009. PMID: 29361717; PMCID: PMC5874548.
  • Hirvonen J, Salminen O, Vuorensola K, Katajavuori N, Huhtala H, Atkinson J. Pharmacy Practice and Education in Finland. Pharmacy (Basel). 2019 Feb 23;7(1):21. doi: 10.3390/pharmacy7010021. PMID: 30813453; PMCID: PMC6473315.

Training in clinical pharmacy is evolving a lot, which goes along with increasing clinical activities provided by pharmacists in France. So we are convinced that it is very relevant to publish it by highlighting the specificities of the situation in France, namely that there is a national system for the whole country that covers all universities.

We added in discussion section others elements of comparison with others countries and suggestions.

Page 11: “A more precise evaluation, beyond the theoretical program, of the number of hours of clinical pharmacy teaching in each French faculty is still difficult to obtain. In the European study, this point was assessed to be 15 hours of clinical pharmacy teaching per semester, which represents the lowest number of hours among the European countries that responded to the survey in 2018 [8]

Page 12: “We highlight four important points worth promoting and developing for clinical pharmacy teachers and state-decision makers.

  • First, pharmacy students need to receive broader training in ethics and health education, to foster a personalized approach based on the patient rather than on the medicine [40], by integrating a preventive approach. Although training in communication and soft skills is now an integral part of French pharmacy studies, the COVID 19 pandemic, has caused the general public to mistrust drugs and pharmaceutical companies, and has been a reminder of the importance of understanding patients’ socio-behavioral features when using medication, to achieve optimal clinical and humanistic outcomes. Social pharmacy needs further developing in pharmacy studies, as the interaction between patients and pharmacists is increasing, notably with clinical pharmacy activities (such as drug reconciliation, pharmaceutical interviews, and therapeutic education…). State decision-makers must display clear public health priorities, with human and material resources to achieve these objectives.
  • Secondly, as in United States [41], declining interest in pharmacy as a career has been observed in France. One of the counteracting levers rests on developing a strong professional identity, also highlighted in a French context [33]. Recently, the 2023–2024 report by the Academic Affairs Standing Committee, published in the American Journal of Pharmaceutical Education, stresses the urge for pharmacy education to prioritize enhancing a stronger professional identity so as to develop collaboration among pharmacists [39]. The major issue that could modify clinical pharmacist practice is to get the pharmaceutical profession to appropriate a caregiver identity [33]. This point must integrate interprofessional education, which is a prerequisite to building a collaborative practice environment and optimizing patient care [42]. Clear guidelines must be proposed in the French faculties of Health to implement this approach more widely.
  • Thirdly, as already mentioned by Planus et al., in 2008, few French researchers have described teaching strategies in pharmacy training [23]. Just as evidence-based medicine is taught to pharmacy students, evidence-based pharmacy education must be developed by pharmacy teachers. Teachers’ involvement in educational innovation should be promoted [24]. Clinical pharmacy training needs to use a combination of teaching techniques, including e-teaching, and to prioritize one-on-one and one-site teaching, using simulated or real-life professional situations to work on soft skills (Figure 3). To develop these approaches, the number of Clinical Pharmacy teachers and their qualification in pedagogical methods must be increased.
  • Fourthly, student assessment must be suited to teaching methods and skills acquisition, and based on objective learning progress assessment [23]. In clinical pharmacy, examinations should allow teachers to assess students' clinical knowledge, communication and problem-solving skills. Objective structured clinical examination formats, gold-standards for evaluating clinical skills in medicine [43], and examinations in real-life professional situations [44] should probably be more widely used by French teachers. Pharmacy faculty members therefore need to increase training in the proper design of educational research [45]. One of digital technology opportunities is that it potentially makes it easier to expand exchanges and bring together French-speaking teachers and students around clinical pharmacy training and educational research. Belgian and Swiss pharmacy teachers recently published an open randomized controlled study comparing an online text-based scenario and a “serious game” to teach triage in the case of coughing. They showed that an online lesson, based on a case study, can be introduced in different countries with only minor changes (e.g., adapting the local drug names)[25]. The French-speaking world offers a potential to develop interrelations and enrich clinical pharmacy education and research”

Reviewer 4 Report

Comments and Suggestions for Authors

Thank you for the opportunity to read the revised version of the manuscript. I think that it has been substantially improved. The aim is clearer and the method is described more in detail. The discussion is also improved.

I have some minor comments.

Abstract: please include that it is a narrative review (line 29)

Introduction: please include the abbreviation ECTS (line 66) since the abbreviation is used later in the manuscript

Table 1: what does internat mean?

Results (line 279): please clarify that the 6 articles included (of 694) were the once that met the inclusion criteria in the method

Line 290: is the correct reference used? Should it be [30] instead?

Table 2 is missing in the manuscript

Author Response

Reviewer 4

The paper describes clinical pharmacy in France and also teaching methods in clinical pharmacy education. Although, a nice description I do not think that the paper is scientific enough and can not be qualified as a review. The methods section is only briefly described and lacks a systematic approach, No attempt is made to for example compare the situation in other countries or put the results in a broader perspective. 

Thank you very much for your time and appreciation of this manuscript.

As you suggested, we removed the term “review” of the manuscript.

We added in discussion section others elements of comparison with others countries and suggestions.

Page 11: “A more precise evaluation, beyond the theoretical program, of the number of hours of clinical pharmacy teaching in each French faculty is still difficult to obtain. In the European study, this point was assessed to be 15 hours of clinical pharmacy teaching per semester, which represents the lowest number of hours among the European countries that responded to the survey in 2018 [8]

Page 12: “We highlight four important points worth promoting and developing for clinical pharmacy teachers and state-decision makers.

  • First, pharmacy students need to receive broader training in ethics and health education, to foster a personalized approach based on the patient rather than on the medicine [40], by integrating a preventive approach. Although training in communication and soft skills is now an integral part of French pharmacy studies, the COVID 19 pandemic, has caused the general public to mistrust drugs and pharmaceutical companies, and has been a reminder of the importance of understanding patients’ socio-behavioral features when using medication, to achieve optimal clinical and humanistic outcomes. Social pharmacy needs further developing in pharmacy studies, as the interaction between patients and pharmacists is increasing, notably with clinical pharmacy activities (such as drug reconciliation, pharmaceutical interviews, and therapeutic education…). State decision-makers must display clear public health priorities, with human and material resources to achieve these objectives.
  • Secondly, as in United States [41], declining interest in pharmacy as a career has been observed in France. One of the counteracting levers rests on developing a strong professional identity, also highlighted in a French context [33]. Recently, the 2023–2024 report by the Academic Affairs Standing Committee, published in the American Journal of Pharmaceutical Education, stresses the urge for pharmacy education to prioritize enhancing a stronger professional identity so as to develop collaboration among pharmacists [39]. The major issue that could modify clinical pharmacist practice is to get the pharmaceutical profession to appropriate a caregiver identity [33]. This point must integrate interprofessional education, which is a prerequisite to building a collaborative practice environment and optimizing patient care [42]. Clear guidelines must be proposed in the French faculties of Health to implement this approach more widely.
  • Thirdly, as already mentioned by Planus et al., in 2008, few French researchers have described teaching strategies in pharmacy training [23]. Just as evidence-based medicine is taught to pharmacy students, evidence-based pharmacy education must be developed by pharmacy teachers. Teachers’ involvement in educational innovation should be promoted [24]. Clinical pharmacy training needs to use a combination of teaching techniques, including e-teaching, and to prioritize one-on-one and one-site teaching, using simulated or real-life professional situations to work on soft skills (Figure 3). To develop these approaches, the number of Clinical Pharmacy teachers and their qualification in pedagogical methods must be increased.
  • Fourthly, student assessment must be suited to teaching methods and skills acquisition, and based on objective learning progress assessment [23]. In clinical pharmacy, examinations should allow teachers to assess students' clinical knowledge, communication and problem-solving skills. Objective structured clinical examination formats, gold-standards for evaluating clinical skills in medicine [43], and examinations in real-life professional situations [44] should probably be more widely used by French teachers. Pharmacy faculty members therefore need to increase training in the proper design of educational research [45]. One of digital technology opportunities is that it potentially makes it easier to expand exchanges and bring together French-speaking teachers and students around clinical pharmacy training and educational research. Belgian and Swiss pharmacy teachers recently published an open randomized controlled study comparing an online text-based scenario and a “serious game” to teach triage in the case of coughing. They showed that an online lesson, based on a case study, can be introduced in different countries with only minor changes (e.g., adapting the local drug names)[25]. The French-speaking world offers a potential to develop interrelations and enrich clinical pharmacy education and research”
